# Energy Efficiency as a Wicked Problem

**Patrik Thollander [1,2,\*], Jenny Palm [3] and Johan Hedbrant [4]**

[1] Division of Energy Systems, Department of Management and Engineering, Linköping University, 581 83 Linköping, Sweden

[2] Department of Building, Energy and Environment Engineering, University of Gävle, 801 76 Gävle, Sweden

[3] IIIEE, International Institute for Industrial Environmental Economics, Lund University, 223 50 Lund, Sweden; jenny.palm@iiiee.lu.se

[4] Division of Applied Thermodynamics and Fluid Mechanics, Department of Management and Engineering, Linköping University, 581 83 Linköping, Sweden; johan.hedbrant@liu.se

\* Correspondence: patrik.thollander@liu.se

**Abstract:** Together with increased shares of renewable energy supply, improved energy efficiency is the foremost means of mitigating climate change. However, the energy efficiency potential is far from being realized, which is commonly explained by the existence of various barriers to energy efficiency. Initially mentioned by Churchman, the term "wicked problems" became established in the 1970s, meaning a kind of problem that has a resistance to resolution because of incomplete, contradictory, or changing requirements. In the academic literature, wicked problems have later served as a critical model in the understanding of various challenges related to society, such as for example climate change mitigation. This aim of this paper is to analyze how the perspective of wicked problems can contribute to an enhanced understanding of improved energy efficiency. The paper draws examples from the manufacturing sector. Results indicate that standalone technology improvements as well as energy management and energy policy programs giving emphasis to standalone technology improvements may not represent a stronger form of a wicked problem as such. Rather, it seems to be the actual decision-making process involving values among the decision makers as well as the level of needed knowledge involved in decision-making that give rise to the "wickedness". The analysis shows that wicked problems arise in socio-technical settings involving several components such as technology, systems, institutions, and people, which make post-normal science a needed approach.

**Keywords:** energy policy; energy management; wicked problem; energy management system; industry; energy efficiency

## 1. Introduction

The United Nation's 2030 Agenda for Sustainable Development emphasizes that present needs are to be met without affecting future needs in a negative way. From this perspective, improving energy usage is of vital importance. Together with increased shares of renewable energy supply, improved energy efficiency is an obvious way to mitigate climate change. Despite decades of energy-efficiency work, potential remains exist due to various barriers [1,2].

Examples of such barriers may be information asymmetries and imperfections, bounded rationality, power, and culture [1,3]. Research in the area has mainly revolved around the implementation of cost-effective energy-efficiency technologies. One critique to the current research approaches is the large focus on information [4], while for example Thollander and Palm [5] emphasized the importance of considering culture, routines, and values for how energy efficiency is approached in an organization. Moreover, earlier research has stated that there is a large untapped potential for non-technology operational improvement measures resulting from sound

energy management principles [6,7]. Another line of argument is that there has been a lack of a systems approach in energy efficiency research [8]. One perspective that could contribute to an enhanced understanding of improved energy efficiency is the perspective of "wicked problems". A wicked problem is a problem that has a resistance to resolution because of incomplete, contradictory, or changing requirements. The solution depends on how the problem is formulated, and vice versa, the problem definition depends on the ideas about the solution. Efforts to solve a wicked problem may impair the situation or create other problems because of complex interdependencies. Relating to Valentine et al. [9], wicked problems arise when different worldviews collide.

One example can be derived from the Swedish industrial energy policy program PFE (Program for improving energy efficiency in energy intensive industry). Stenqvist and Nilsson [10] could prove that the policy instrument as such was effective, and from a technical perspective, large savings have occurred for the two five-year program periods. This indicated a "success" or "effectiveness" of the policy, but in a governmental report, also bringing in the neoclassical economic perspective on the policy, the improvements could also be due to other mechanisms that are not necessarily connected to the PFE [11].

A second example of a wicked problem is the view of what is and what is not an energy efficiency measure. Relating again to Valentine et al. [9] who stated that " . . . a high degree of technological entrenchment when it comes to decision-making in the energy sector and commitment to certain technologies blinds decision-makers when it comes to evaluating alternative technologies . . . illustrates that special interests and ideologies shape our interpretation of even the most uncontentious of facts". Recent research of the above-mentioned Swedish policy program PFE showed that a large degree of the measures was in fact not directly related to technology, but instead were found in the management and operations of technologies and processes [12]. The PFE program was the first international energy policy for industry to execute a standardized energy management system in the policy context, i.e., prior to the launch of the international energy management standard. Also, most measures were not standalone technology measures, but rather related to system improvements, which demands extensive in-house process knowledge [8].

Below, we will describe the primary method followed by a literature review of wicked problems. Thereafter, the reviewed articles are related to the characteristics of wicked problems used by Rittel and Webber [13]. Then, we discuss earlier research in industrial energy efficiency from a wicked problem perspective.

## 2. Materials and Methods

The major method in our analysis was a literature review. We searched for studies that have used the wicked problem framework to analyze an energy-related empirical phenomena. Only studies explicitly using the term wicked problem were of interest, to give understanding of the use of this term in energy-related studies. The database that was used for the literature search was Web of Science. This database includes the major journals in the energy field and is currently ranked as the most renowned in terms of quality, since it only includes journals and conferences with a sound peer-review process. Papers were searched for from the year 1975. The search results were as follows:

- "wicked problem" and "energy management"—10 papers
- "wicked problem" and "energy"—one paper
- "wicked" and "energy"—60 papers, where 32 were found relevant
- Screening of reference lists gave another five papers
- In the end, 19 papers were suitable for our analysis and are described more in detail.

In the next section, we discuss the reviewed articles in relation to Rittel and Webber's [13] 10 characteristics of a wicked problem.

### 3. Wicked Problems and the Articles Discussing Them

When discussing the found articles on wicked problems in the energy area, the structure will follow Rittel and Webber's [13] 10 characteristics. Most papers relate to several of them, but each paper below will only be discussed in relation to the most contributing characteristic.

Wicked problems were first outlined by Churchman [14] referring to Horst Rittel. Churchman described a wicked problem as a social system problem that is usually not well formulated and with a confusing setting. The solutions generally include several decision makers with conflicting values, and occasionally, the proposed solution to a problem was worse than the symptoms. Some years later, Rittel and Webber [13] wrote an article that specified 10 characteristics of wicked problems in social policy planning. We will discuss each of the characteristics below in the order that they appear in Rittel and Webber [13]. The headings are Rittel and Webber's formulation.

1. There is no definitive formulation of a wicked problem.

*The information needed to understand the problem depends upon one's idea for solving it. The problems of formulating the problem and conceiving a solution are identical. Every question asked will depend on the understanding of the problem and its resolution at that time. To anticipate all of the questions, knowledge of all possible solutions is needed in order to anticipate all of the information required for resolution. In order to describe a wicked problem in sufficient detail, an exhaustive inventory of all possible solutions is needed [13].*

The study by Laws, Hogendoorn, and Karl [15] analyzed the impact of conflicting views in relation to adaptive co-management of social–ecological systems. They studied earlier research and discussed the possibilities of working collaboratively under conditions of conflict. Due to the uncertainty and ambiguity that wicked problems involve, technical analysis is unlikely to provide a final resolution. Therefore, a first step is a deliberative process where stakeholders learn about the problem together, a so-called joint fact-finding. Then, a battle can lead to a constructive dialogue. This can lead to a situation where the focus is to get on with the situation rather than discuss whose assertions are true.

2. Wicked problems have no stopping rule.

*In solving a chess problem or a mathematical equation, the problem-solver knows when he has done his job. Not so with a wicked problem. There are no criteria for sufficient understanding and no ends to the causal chains that link interacting open systems. There are always chances to find a better solution [13].*

This is discussed in a paper by Everingham et al. [16]. Using semi-structured interviews, Everingham et al. conducted a study in Australia of conflicts between coal mining, gas extraction, and land use related to agriculture. The paper summarizes the major characteristics related to wicked problems and finds, among other things, that increased amounts of scientific information has led to an increase in uncertainty in actually ever solving the problems.

3. Solutions to wicked problems are not "true or false", but rather "good or bad".

*For wicked problems there are no conventionalized criteria for objectively deciding whether the offered solution is correct or false. Rather, it can be "good or bad", "better or worse", or "good enough" [13].*

Turnpenny et al. [17] in their article discussed post-normal science (PNS) as a way to handle wicked problems. They did a documentary analysis of three cases that they argued were wicked in nature. The cases are Canadian regulations of health products and food, European Union environmental thematic strategies, and United Kingdom energy and climate change policy. They showed that wicked problems are approached very differently in the three cases. Participation was embedded in all of the cases, but political interests directed the course of action. They saw a potential for PNS, but at the same time, the authors called for a better definition of PNS and further research on how to integrate it into other branches of social science.

Also, Fast and McCormick [18] emphasized the role of PNS to enroll more perspectives and more thinking styles to work with wicked problems. Fast and McCormick assessed experiences of biofuels in the European Union (EU) and United States (US) against the 10 characteristics of wicked problems

and argued that biofuels fit the profile of a wicked issue. One conclusion is that transport biofuels need recognition and the engagement of multiple perspectives; therefore, they advocated for having a PNS approach to the issue.

4. There is no immediate and no ultimate test of a solution to a wicked problem.

*The full consequences of a solution to a wicked problem cannot be appraised until the waves of repercussions have completely run out, and there is no way to trace all of the waves through all the affected lives ahead of time [13].*

This rule is not really touched upon in the found papers, but two of them discussed tools or methods to support decision-making or at least test how well a decision can meet a wicked problem. Bhave et al. [19] studied Robust Decision Making (RDM) approaches among developing countries and stated that RDM can be a means to support parts of the decision-making regarding the "super-wicked" problem of climate change.

Janda et al. [20] found that the improvements of non-domestic buildings represented a wicked problem, in particular the split incentive and landlord–tenant dilemma and suggested that green leases may be one tool to address challenges in relation to this.

5. Every solution to a wicked problem is a "one-shot operation" because there is no opportunity to learn by trial and error, every attempt counts significantly.

*With wicked problems every implemented solution is consequential and leaves traces that cannot be undone [13].*

Viganò's [21] article focused on places where climate change-induced hazards will be especially important. He used the lagoon location of Venice and the Garonne riverbank location of Bordeaux as examples where rising sea level will affect the urbanized coast as well as riverbanks and lagoons. The article discussed different adaptation strategies to increased disaster risks. Viganò argued that urban and territorial design have to integrate the inevitability of risk as a way to deal with wicked problems.

6. Wicked problems do not have an enumerable (or an exhaustively describable) set of potential solutions, nor is there a well-described set of permissible operations that may be incorporated into the plan.

*There exist no criteria which enable one to prove that all solutions to the problem have been identified and considered. All ideas are worth trying out [13].*

Klinzing (2010) discussed systems for pneumatic transport and wicked problems, and stated that, e.g., due to electrostatics, pneumatic transport systems can face unplanned shutdowns that are not at all possible to anticipate. However, Klinzing [22] argued that wicked problems can be 'tamed', and discussed strategies and previous methods to do so. Most of the methods involved getting expert teams or groups together to begin to communicate, discuss, and brainstorm.

Irwin [23] did not discuss taming wicked problems, but his reasoning was similar. Irwin wanted to solve wicked problems such as climate change with transition design. Transition design will imply a reconception of the entire lifestyle and a reimagination of infrastructures, which include energy as well as healthcare and education. Transition design focuses on "cosmopolitan localism", where a change in lifestyle takes place locally, but with a global awareness. However, the idea is that of letting many flowers flower.

7. Every wicked problem is essentially unique.

*There are no classes of wicked problems, and thus there is not a solution that fits all the members of a class. Despite long lists of similarities between current problems, there are always distinguishing properties of overriding importance [13].*

Totten [24] discussed that business as usual is trespassing planet boundaries in the terrestrial, atmospheric, and oceanic spheres. There are multiple wicked problems that need to be solved in our time. However, Totten argued that one solution is to leverage the power of self-organizing small-world

networks of cooperation among willing citizens. The wicked problems should be solved at a local scale, and Totten argued that urban and rural landscapes could achieve zero-emissions targets.

This bottom–up approach is supported by Ardoin et al. [25], who conducted an interview study among environmental leaders on how to face wicked problems in environmental decision-making and also ended up promoting localized problem solving.

8. Every wicked problem can be considered to be a symptom of another problem.

*Problems can be described as discrepancies between the state of affairs as it is and the state as it ought to be. The process of resolving the problem starts with the search for a causal explanation of the discrepancy. Removal of that cause poses another problem of which the original problem is a "symptom" [13].*

Bakhsi's [26] paper is very much in line with this statement. He wrote a reflective paper based on earlier engineering studies on sustainability among businesses, and argued that used methods such as life cycle assessments as well as the focus on enhancing the efficiency of resource and energy use for a process are not enough. Since sustainability is a wicked problem, additional efforts are required. These include taking into account the role of ecosystem services in life cycle analysis and decision-making, and encouraging strong multidisciplinarity when solving these issues.

Also, Bouma and McBratney [27] focused on the interlinkage of problems, and used soil degradation and management as an example. They argued that environmental problems such as energy, water, and food security are wicked problems that are all connected to soil management. The solution is to establish interdisciplinary teams to study these "wicked" environmental problems, which would assure that soils are included in the analysis right from the start.

The systemic character of a wicked problem was stressed also by Azapagic and Perdan [28]. They studied the supply of energy and biofuels within the chemical engineering field and stressed that a systems approach is a must in order to battle the wicked problems within this area. They argued that a systems approach should include working with other disciplines, e.g., humanities and social sciences.

9. The existence of a discrepancy representing a wicked problem can be explained in numerous ways.

*The choice of explanation determines the nature of the problem's resolution. A wicked problem can be explained in different ways, which all offer directions for attacking the problem. There is no correct way or rule to decide which explanation or combination of them is correct [13].*

Barrick [29] discussed this in relation to geysers. More than half of the world's remaining geysers are located in Yellowstone National Park, northwest Wyoming, US. The hydrothermal reservoirs that supply Yellowstone's geysers extend beyond the park borders, and Barrick suggested establishing a Geyser Protection Area. Barrick argued that it is important that a Geyser Protection Area not become a wicked policy conflict where the issue becomes acrimonious, symbolic, and intractable, and goes beyond standard scientific, economic, and techno-rational problem-solving methods. Several of Yellowstone's environmental policy disputes have become wicked in recent years, and Barrick's [29] study indicates that wicked problems are in a way a social construct that is possible to avoid. The need to have a social constructivist view on wicked problems was upheld by Valentine et al. [9]. The paper provides a response to the critique from positivism that social constructivism does not have a role to play in policy analysis. In their conclusions, they drew up six maxims to help improve understandings of wicked energy problems.

10. The planner has no right to be wrong.

*Traditionally, scientific solutions to problems are only hypotheses offered for refutation. However, this is not true for a wicked problem. The aim is not to find the truth, but rather to improve the world in which people live. The wicked problems are incorrigible, and planners are liable for the consequences of the actions that they generate [13].*

City planning as a wicked problem was discussed by Engberg [30] and Yearworth [31] and also partly by Pere and Farrell [32]. Pere and Farrell [32] reviewed the wind energy siting debate in the Catalonia region in Spain over the past 30 years. They argued that wind energy siting can be viewed

as a wicked problem. They analyzed the debate as "one-dimensional thinking" where the debate is confined to the dimension of "facts", the actuality or appearance of things, and debate about the dimension of "values", the potentiality or purpose of things, is suppressed or rejected. The authors argued that this one-dimensional thinking has many similarities with wicked problems.

## 4. Analysis

In this section, we will discuss improved energy efficiency from the perspective of wicked problems. We will go through the 10 characteristics framing a wicked problem and relate each one of them to these. To avoid repetition, some of the 10 proposals will be discussed together.

1. There is no definitive formulation of a wicked problem.

Energy system transition will require investments in more energy-efficient equipment, but there is also a need to "transform" people's attitudes, behaviors, values, and routines, as well as improve the knowledge of specific production systems and processes [6]. Energy management is about reducing costs for the provision of energy in buildings and facilities without compromising the working environment and the production processes, etc. [33]. This is easy in theory but depends on the management's understanding of the problem. There are e.g., technological, economical, or social aspects that the governance or leadership need to consider. Energy efficiency can be framed in financial terms related to future energy prices and the cost of energy-related processes. Investments in energy efficiency can also be ecologically motivated as a concern for the environment in general. It can also be an ethical consideration: the importance of not wasting future generations' resources, or that the company needs to invest due to state policy today and in the future. Depending on how the issue is framed, different parts of a company will be activated. The units and professions in the company will have different approaches to what measures to implement and how. This may give precedence to some perspectives of why the energy use is a problem (compare [5]).

Perhaps the energy efficiency discourse is the problem. According to Shove [34], the problem with energy efficiency is that it involves maintaining the status quo, rather than challenging a present lifestyle. To reach sustainability, we need to reduce the total energy use (or make it carbon-neutral) and not focus on efficiency. Reducing energy use requires far more radical changes in the system. Existing structures and dominant actors (e.g., governmental bodies and other organizations, national as well as international) need to evaluate their own role in making and shaping future needs, and as long as this does not happen, the definition of a sustainable energy system remains unclear [34] and remains a wicked problem.

2. Wicked problems have no stopping rule.

Energy management in the later attempts has been formulated into an international standard, ISO 50001, where "continuous improvements" is one of the fundamentals. This means that there is no clear end or stop to improving energy efficiency. The stopping criteria will depend on the problem definition discussed above, the problem that improved energy efficiency wants to solve. When to stop improving energy efficiency in an industrial setting will also depend on e.g., cost efficiency, competition, production capacity, employees' health and safety, and the knowledge of the specific system and processes in the company.

3. Solutions to wicked problems are not "true or false", but rather "good or bad".

4. There is no immediate and no ultimate test of a solution to a wicked problem.

Improvements of energy efficiency may take fundamentally different forms and approaches depending on the type of production, building stock, management culture, etc. In one company, it may take technological approaches, and in another focus on management [35]. Sometimes, very deep knowledge of energy use in specific production processes is needed [8].

Establishing an in-house energy management program occasionally comes with some level of conflict. The production manager's focus is on productivity and changing machines and routines to improve energy efficiency might be risky from a productivity perspective. The quality manager's focus, on the other hand, is on zero defects, and might oppose the replacement of old reliable equipment, etc. Every company is unique, and every interference can be done in different ways with multiple purposes. Energy efficiency is in this way locally colored, and it is almost always possible to evaluate an implemented measure with the conclusion that it could have been done otherwise [5,36].

5. Every solution to a wicked problem is a "one-shot operation".

6. Wicked problems do not have an enumerable (or an exhaustively describable) set of potential solutions, nor is there a well-described set of permissible operations that may be incorporated into the plan.

In the simplest forms, the example of compressed air system measures [22] provides a good example of how energy management in companies is path dependent. Many companies have utilized the spill heat generated by the compressors. However, the sealing of leaks, pressure reductions, or conversions from pneumatic devices to energy-efficient electric drives may be left unexploited, with the result being that if such actions are undertaken, the newly installed heat exchanger may remain too large in size and end up being a potential waste of company resources.

Up until today, no clear complete strategy exists for how energy management activities or actions could be carried out. Models exists, e.g., Schulze et al. [37], on how in-house energy management is being operationalized. The implicit assumption made in these available models is often that the implementation of an energy management system leads to continuous improvements, or that improved energy efficiency is achieved by the implementation of BAT (Best Available Technologies) [5]. The input–output model is also the major model used within the European Union (EU) energy policy action plan formulation. The underlying logic for the models that is used to motivate both the diffusion of information and the implementation of energy management practices are weakly linked with present theory building within the area of improved energy efficiency. A critical aspect is that the scientific contributions in the field of energy management emanates mostly from researchers based in engineering science. Thus, it is important to further explore the area of improved energy efficiency in terms of the models used, and new models needed, not least from a socio-technical perspective.

7. Every wicked problem is essentially unique.

8. Every wicked problem can be considered to be a symptom of another problem.

In relation to this part of wicked problems, one of the earlier works within industrial energy management may be cited: Caffal [35] outlined that there is no "one size fits all" solution. The same was later stated by Christoffersen et al. [33]. Improved energy efficiency, energy management, and energy efficiency policy programs may all need to be streamlined to fit the specific targeted sector, type of organization, or type of energy efficiency measures.

Thus, this statement seems to not be fully true. Regardless of whether improved energy efficiency, energy management, or energy efficiency policy programs is in focus, the underlying effect or mechanism is that of inefficiency. In this sense, every problem is not unique.

However, if one sees the inefficiency as a "symptom" of a number of different factors, each one contextual, e.g., lack of staff motivation, lack of knowledge on how to procure a more energy-efficient technology, information asymmetries and imperfections, to name a few, this statement may be regarded as valid. If a very simple form of energy efficiency improvement, such as a consumer replacing an old incandescent lamp with a light-emitting diode (LED) lamp, this improvement may not be regarded as a wicked problem. However, the more one moves into additional complex improvement measures, the attempt also involves aspects such as procurement, the relationship between the buyer and the seller, in-house energy management programs demanding extensive contextual know-how about the

specific energy end-using technologies, processes, and their interaction, then the problem seems more similar to a wicked problem.

9. The existence of a discrepancy representing a wicked problem can be explained in numerous ways.

10. The planner has no right to be wrong.

Closely related to the distinction between energy management operations, setting up routines, creating a baseline, etc., and that of technology improvements, one may improve efficiency from very different perspectives. As stated by Caffal [35], the 40% improvements that the author outlined were a combination of management approaches and technology improvements. The same holds for energy efficiency policy programs where the most straightforward policy approach is energy audit policy programs, while some countries, such as Japan, Sweden, and Belgium, have had ongoing Voluntary Agreement Programs as major policy approaches toward energy-intensive industry [38]. Thus, there is no true or false means for achieving improvements on technological, process, management, or policy levels.

A summary of energy efficiency as a wicked problem is found in Table 1.

**Table 1.** Summary of how improved energy efficiency can be understood as a wicked problem.

| The 10 Characterizations of Wicked Problems | Relation to Energy Efficiency |
|---|---|
| There is no definitive formulation of a wicked problem. | Energy efficiency can be framed as a techno-economic problem, or a problem related to fundamental issues as lifestyle, behavior, values, and routines. |
| Wicked problems have no stopping rule. | Energy efficiency is a continuous process and must be related to many different factors such as cost efficiency and employees' health and safety. |
| Solutions to wicked problems are not "true or false", but rather "good or bad", and there is no immediate and no ultimate test of a solution to a wicked problem. | Energy efficiency changes form in different management cultures and in different sectors. "Waste of energy" is an expression in some sectors, while not at all discussed in others. |
| Every solution to a wicked problem is a "one-shot operation" and wicked problems do not have an enumerable (or an exhaustively describable) set of potential solutions, nor is there a well-described set of permissible operations that may be incorporated into the plan. | Improved energy efficiency needs to be treated as a technological problem as well as a management and behavioral problem that can be approached in many different ways. |
| Every wicked problem is essentially unique, and every wicked problem can be considered to be a symptom of another problem. | There is no "one size fits all" solution, but the underlying mechanism is inefficiency, which in turn may be due to several factors. |
| The existence of a discrepancy representing a wicked problem can be explained in numerous ways, and the planner has no right to be wrong. | A lot of different means exist to improve energy efficiency around the world, and they all have the potential to contribute to a solution. |

## 5. Discussion

Energy efficiency can be understood as a wicked problem because it is unclear what actually the problem is when it comes to energy use and improved efficiency. Is it that we already today use too much, and in this perspective should start discussing in terms of sufficiency instead? Or is the problem the more traditional view that we need to improve the efficiency of the usage so that we do not need to increase energy demand when we increase the number of units?

The answer to this probably depends on what societal sector or research community one belongs to, or in other words, which model one has for understanding energy efficiency. Is energy efficiency represented by a diffusion of energy-efficient technologies, is it an effect of energy system improvements, or is it related to energy management improving the operational part of energy use? This relates back to Turnpenney et al. [17] and Fast and McCormick's [18] discussion on the need for post-normal science to have an ongoing discussion between perspectives and knowledge types on how to perceive energy efficiency.

It can be concluded that improved energy efficiency involving only technical installations as such would not be enough for defining it as a wicked problem. However, Klinzing [22] meant that even in such cases, technical energy efficiency improvement measures may be seen as wicked. However, in

a later paper, Klinzing [39] left the term "wicked problem" and instead classified these problems as "unusual", which was most likely because there was a sense of agreement on how to define the problem at stake. However, relating to the understanding that improved energy efficiency is understood as technology and process-related improvements or operational measures, these two explanatory models might imply a form of wickedness. It seems to be the actual decision-making process and the level of needed knowledge involved in decision-making that give rise to the "wickedness".

Another reflection from the papers reviewed is that it is not the technology or process as such that is wicked, but rather the actors' ways of discussing and understanding it. Hence, technical aspects are in many cases important, but the elusiveness or wickedness seems to be at another complexity level, including social interactions, meaning, and routines (compare [40]). Further revolving around this topic, sometimes technology as such is not what give rise to improved energy efficiency, but rather energy management and internal deep knowledge of the production process and its related energy use.

As regards operational energy efficiency improvement measures related to the management of technologies and processes and its interaction, the very same conclusion may be drawn. In the very narrow sense of energy management, for an issue such as buying new standalone technologies, no wicked problem seems present. However, there are specific problems, such as the inappropriate shutdown of equipment, that might take the character of a wicked problem, where a common problem definition does not exist and it is not possible to reach a final solution. However, energy management in a broader sense includes the implementation of production process changes that are influencing production lines, organizational culture, behavior, and routines, which in turn may be understood as a wicked problem where it is not possible to agree on one problem definition or what the solution should be.

Standalone technology improvements as well as energy management and energy policy programs giving emphasis to standalone technology improvements may not represent a stronger form of a wicked problem as such. However, if one understands improved energy efficiency as technology-related and process-related improvements as well as operational measures, demanding deep process knowledge and/or relationships with a number of people involved in the improvement process, energy efficiency, energy management, and energy policy programs are indeed strong forms of wicked problems. It seems to be the actual decision-making process involving values among the decision makers as well as the level of needed knowledge involved in decision-making that give rise to the "wickedness".

If improved energy efficiency is to reach its full potential, further research is needed in this important field. This means from the technical perspective that technology and process-related measures as well as how processes and technologies are operated is included. Moreover, this means from the user or decision-maker's perspective that both leadership issues on how to manage staff as well as their own operations and use of energy, as well as deep process knowledge and personal values, are to be included. Only through doing so could an enhanced understanding beyond the understanding of energy efficiency as the only deployment of standalone technologies be reached.

**Author Contributions:** Conceptualization, J.P., P.T., and J.H.; Data curation, J.P. and P.T.; Formal analysis, J.P., P.T., and J.H.; Investigation, J.P., P.T., and J.H.; Methodology, J.P. and P.T.; Project administration, J.P., P.T., and J.H.; Writing—original draft, J.P., P.T., and J.H.; Writing—review and editing, J.P., P.T., and J.H.

**Funding:** This research received no external funding.

**Conflicts of Interest:** The authors declare no conflict of interest.

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
