# Peer review of "Energy Efficiency as a Wicked Problem"

_sustainability, doi:10.3390/su11061569_

Reviewer 1 Report

The article contains excerpts from the guide for authors - this is probably negligence!

The presented problem is current and interesting, but the way it is presented in the article is misleading.

Taking into account the review of literature made by the authors, the conclusion is that we are talking not so much about 10 characteristics and a wicked problem (which is presented in Chapter 3), but rather about ways to optimize the actions taken to improve energy efficiency.

It seems that the title of the article does not correlate with the discussion and conclusions drawn by the Authors because the entire review of the literature and the presented characteristics and the wicked problem have no point at all in the Discussion to improve industrial energy efficiency and achieving energy conservation.

The potential and improvement of energy efficiency is a process, an aspiration and as such is subject to the activities of optimization, modernization, rationalization, and improvement. Some of the results presented in the paper reflect poorly made and ill-chosen ways of achieving, improving energy efficiency. In addition, not all conflicts and contradictory arguments define a wicked problem.

The article does not touch the soft social problems that hinder the process of energy saving and energy justice.

The article requires a careful review of the authors.

Specify the purpose and the research problem and provide clear arguments justifying the occurrence of the wicked problem and not the lack of system optimization. The content of the article does not relate to industrial energy efficiency and achieving energy conservation - it requires changes.

Author Response

Reviewer 1

The article contains excerpts from the guide for authors - this is probably negligence!

RESPONSE: Thanks, we sincerely apologize for this mistake!

The presented problem is current and interesting, but the way it is presented in the article is misleading.

Taking into account the review of literature made by the authors, the conclusion is that we are talking not so much about 10 characteristics and a wicked problem (which is presented in Chapter 3), but rather about ways to optimize the actions taken to improve energy efficiency.

It seems that the title of the article does not correlate with the discussion and conclusions drawn by the Authors because the entire review of the literature and the presented characteristics and the wicked problem have no point at all in the Discussion to improve industrial energy efficiency and achieving energy conservation.

The potential and improvement of energy efficiency is a process, an aspiration and as such is subject to the activities of optimization, modernization, rationalization, and improvement. Some of the results presented in the paper reflect poorly made and ill-chosen ways of achieving, improving energy efficiency. In addition, not all conflicts and contradictory arguments define a wicked problem.

The article does not touch the soft social problems that hinder the process of energy saving and energy justice.

The article requires a careful review of the authors.

RESPONSE: Thank you for this up-right and honest reflection. Following the request from the second reviewer, we have introduced more bullet points and tables, and shortened the text, i.e. to make it more concise. We hope this has also clarified how we see the connection to the wicked problems. We also agree on that we had more focus on system then on wickedness in the first version, and have therefore tried to revise also this, in this revised version. Thanks again for your comments.

Specify the purpose and the research problem and provide clear arguments justifying the occurrence of the wicked problem and not the lack of system optimization. The content of the article does not relate to industrial energy efficiency and achieving energy conservation - it requires changes.

RESPONSE: Tanks again. We have clarified the aim and changed the focus from the system aspect towards wickedness. We think that the article relates as you write, more on energy efficiency in general. We have therefore removed “industry” in some parts of the paper. Also, we have deleted energy conservation.

Reviewer 2 Report

Clearly objectives of the study, methoology and expected results has to be declared (e.g in bullet points)

The article has big repetitons, same period is repeated in different paragraphs (see file).

Improve english is necessary. I suggest to cut the text, that is very long, moslty in the chapter 3, and to introduce some synoptic tables, or image. At the moment it resuts difficlut to read.

Also conclusions and discussion have to be summarized or be more clearly explained. Bulett points or evidences. 

Author Response

Reviewer 2

Clearly objectives of the study, methoology and expected results has to be declared (e.g in bullet points)

RESPONSE: Thank you for this suggestion. We have clarified the aim and included more bullet points.

The article has big repetitions, same period is repeated in different paragraphs (see file).

RESPONSE: Thank you for all comments in the paper. We have corrected this in accordance to your comments.

Improve English is necessary. I suggest to cut the text, that is very long, mostly in the chapter 3, and to introduce some synoptic tables, or image. At the moment it results difficult to read.

RESPONSE: Thank you, we have proof-read the paper. We have also tried to reduce the text and introduced more bullet points and tables, in order to make the paper more concise.

Also conclusions and discussion have to be summarized or be more clearly explained. Bullet points or evidences. 

RESPONSE: Thanks for your comment. We have shortened the conclusions and tried to write them more spot on.

Round  2

Reviewer 2 Report

paper is now acceptable. Overall objectives of the study are clear and conclusions are appropriate. English has been improved